# Spatial Transcriptomics in Thyroid Cancer: Applications, Limitations, and Future Perspectives

**DOI:** 10.3390/cells14120936

**Published:** 2025-06-19

**Authors:** Chaerim Song, Hye-Ji Park, Man S. Kim

**Affiliations:** 1Translational-Transdisciplinary Research Center, Clinical Research Institute, Kyung Hee University Hospital at Gangdong, College of Medicine, Kyung Hee University, Seoul 05278, Republic of Korea; chaerim1052@gmail.com; 2Department of Oral Medicine, Kyung Hee University Hospital at Gangdong, Seoul 05278, Republic of Korea; hyeji@khnmc.or.kr

**Keywords:** spatial transcriptomics, thyroid cancer, tumor microenvironment, precision medicine

## Abstract

Spatial transcriptomics (ST) is emerging as a powerful technology that transforms our understanding of thyroid cancer by offering a spatial context of gene expression within the tumor tissue. In this review, we synthesize the recent applications of ST in thyroid cancer research, with a particular focus on the heterogeneity of the tumor microenvironment, tumor evolution, and cellular interactions. Studies have leveraged the spatial information provided by ST to map distinct cell types and expression patterns of genes and pathways across the different regions of thyroid cancer samples. The spatial context also allows a closer examination of invasion and metastasis, especially through the dysregulation at the tumor leading edge. Additionally, signaling pathways are inferred at a more accurate level through the spatial proximity of ligands and receptors. We also discuss the limitations that need to be overcome, including technical limitations like low resolution and sequencing depth, the need for high-quality samples, and complex data handling processes, and suggest future directions for a wider and more efficient application of ST in advancing personalized treatment of thyroid cancer.

## 1. Introduction

Thyroid cancer is the most common endocrine malignancy, with its global incidence rate continuously rising over the years [1]. It is approximately four times more common in women than men, with the incidence rate peaking for women at 40–49 years [2]. Histological subtypes of follicular cell-derived thyroid cancers include papillary thyroid carcinoma (PTC), follicular thyroid carcinoma (FTC), poorly differentiated thyroid carcinoma (PDTC), and anaplastic thyroid carcinoma (ATC). PTC accounts for approximately 80% of the cases and generally has a favorable prognosis with a survival rate greater than 90% [3]. ATC, on the other hand, exhibits more aggressive trajectories accompanied by metastasis to lymph nodes and distant organs [4]. Some treatment options for thyroid cancer include surgery (thyroidectomy or lobectomy), radioactive iodine therapy, thyroid hormone therapy using levothyroxine, and molecular-targeted therapies using tyrosine kinase inhibitors [5,6,7,8]. Immunotherapy options, such as immune checkpoint blockade, cancer vaccines, and tumor-associated macrophage targeting, are also on the rise, especially for ATC which is often resistant to traditional treatments [9].

These current therapies, however, lack the precision to target specific mutations and pathways and are limited in addressing variable clinical responses due to tumor hetereogeneity. This has highlighted the need for more advanced technologies to better characterize tumors at the molecular level. Next-generation sequencing technologies offer a powerful means to uncover the genomic and transcriptomic alterations driving disease progression, revolutionizing cancer research towards a more personalized diagnosis and treatment for patients. RNA sequencing (RNA-seq), in particular, has provided valuable information into the diversity and function of thyroid cancer by offering profound insights into the differences in gene expression. For example, Pu et al. identified subpopulations of malignant thyrocytes and explored their potential in immunotherapy by analyzing single-cell RNA-seq (scRNA-seq) data from PTC patients [10]. Multiple studies integrating bulk RNA-seq and scRNA-seq have also elucidated intricate molecular mechanisms of the cancer, such as the critical roles of CADM1–CADM1 interactions in the regulation of PTC or the high infiltration of fibroblasts in PTC [11,12].

However, both bulk RNA-seq and scRNA-seq fail to capture the spatial context of tissues, limiting our understanding of the tissue architecture and impeding precise targeted therapies. To overcome such an impediment, spatial transcriptomics (ST) has recently emerged as a powerful approach in providing spatial coordinates within a tissue, allowing researchers to map gene expressions to specific locations. Unlike traditional sequencing methods that require cell dissociation, it captures mRNA molecules directly from either slides or beads containing spatial barcodes [13,14,15]. Further, it provides critical insights into the regional differences across the unique microenvironment of each tumor, such as tissue architecture, cellular organization, dynamic cellular interactions, and patterns of dedifferentiation and immune infiltration that may be missed with conventional study methods. ST thus paves the way for more accurate diagnoses and personalized treatment plans tailored to the unique spatial characteristics of a patient’s tumor microenvironment (TME). The technology is particularly valuable for studying solid tumors like thyroid cancer, where the spatial arrangement of malignant cells, immune infiltrates, stromal cells, and vasculature plays a critical role in disease progression, therapeutic response, and resistance. The histologically complex and heterogeneous nature of thyroid cancer tissues also underscores the need for preserving the spatial information of the cells or tissue regions within the tumor [16].

This review aims to discuss the potential of ST in thyroid cancer research by exploring its recent contributions to the field, challenges to overcome, and future directions in further advancing its applications. Overall, we hope to foster a deeper understanding of this burgeoning technology to optimize more personalized diagnosis and treatment strategies for thyroid cancer.

## 2. Overview of ST

ST is an innovative class of technologies that enables the measurement of gene expressions while retaining the spatial organization of cells within intact tissue sections. Unlike bulk RNA-seq, which averages gene expressions across heterogeneous cell populations, or scRNA-seq, which requires tissue dissociation and sacrifices spatial context, ST preserves the tissue architecture and allows researchers to map transcripts back to their original histological locations. By integrating molecular profiling with spatial context, ST bridges the gap between histopathology and transcriptomics, allowing a range of functions like dissecting tumor heterogeneity, mapping tumor–immune interactions, and identifying niche-specific drug resistance mechanisms. To help contextualize how ST is applied in thyroid cancer research, we first briefly provide an overview of the workflow of ST and the common platforms used (Figure 1).

### 2.1. Workflow and Methodology of ST

ST begins by preparing tissue sections, typically fresh-frozen or formalin-fixed paraffin-embedded (FFPE), mounted onto specialized slides containing spatially barcoded capture probes. Tissue sections are stained, commonly with hematoxylin and eosin (H&E) or immunofluorescence, for histological reference, followed by high-resolution imaging. This image serves as a spatial map to which transcriptomic data are later aligned.

After imaging, the tissue is permeabilized to release mRNA molecules, which then hybridize with the barcoded oligonucleotides in a position-dependent manner. These probes include a spatial barcode unique to each capture spot or bead, a poly(dT) sequence to bind polyadenylated transcripts, and a unique molecular identifier for transcript quantification. Captured transcripts are then reverse-transcribed, amplified, and subjected to high-throughput sequencing. The resulting data are computationally processed to map gene expressions back to precise spatial locations within the tissue [17].

The resolution and sensitivity of ST depend on several factors, including tissue quality, thickness, permeabilization conditions, and RNA integrity. ST data are often integrated with histological features and other data types like proteomics or scRNA-seq to provide a multimodal understanding of tissue architecture and cellular interactions. In the context of thyroid cancer, this workflow enables the spatial mapping of oncogenic gene expression patterns, immune infiltration, and microenvironmental heterogeneity, offering insights beyond what bulk or dissociated single-cell approaches can achieve.

### 2.2. ST Platforms

There are various platforms available for performing ST, each differing in resolution, transcript coverage, and technical requirements. These platforms can be broadly categorized into sequencing-based and imaging-based methods.

Sequencing-based platforms, such as 10× Genomics Visium, Slide-seq, and Slide-seqV2, use arrays or beads embedded with spatially barcoded capture probes to capture mRNA. 10× Genomics Visium, for example, uses spatial capture spots approximately 55 µm in diameter that can cover multiple cells per spot, enabling whole-transcriptome analysis with moderate resolution [18]. Slide-seq and Slide-seqV2 can improve spatial resolution to about 10 µm by using randomly distributed barcoded beads on a slide, enabling near single-cell resolution [15]. These platforms offer unbiased transcriptome-wide coverage and are well-suited for fresh-frozen tissues.

Imaging-based platforms, such as Multiplexed Error-Robust FISH (MERFISH), sequential fluorescence in situ hybridization (seqFISH), and CosMx Spatial Molecular Imager (SMI), use rounds of hybridization and high-resolution imaging to detect hundreds to thousands of predefined RNA targets at subcellular resolution [19,20]. Although these methods offer finer spatial resolution than sequencing-based platforms, they are generally restricted to targeted panels and require more complex imaging infrastructure and analysis pipelines [17].

Each platform presents trade-offs in spatial resolution, transcriptomic coverage, scalability, and sample type compatibility. Visium, for instance, is ideal for broad discovery in research settings, whereas CosMx may be more applicable for targeted clinical assays. As ST technology continues to evolve, improvements in resolution, sensitivity, and multimodal integration are making it increasingly accessible and powerful for the clinical research of thyroid cancer.

## 3. Spatial Heterogeneity of the Tumor Microenvironment (TME)

The TME refers to the ecosystem surrounding a tumor, comprising various types of cells like cancer-associated fibroblasts (CAFs), immune cells, extracellular matrix components, blood vessels, and signaling molecules [21]. Due to its pivotal role in tumor initiation and progression, immune responses, metastasis, and therapeutic outcomes, there has been an increasing focus on understanding the various cellular populations and their communications within the environment. The thyroid cancer TME is highly heterogeneous and can be characterized in many different ways, including normal and malignant regions, immune infiltration zones, or the leading edge and the core. ST provides the spatial information of cellular compositions and gene expressions within the TME, allowing the mapping of different cell types, the identification of subpopulations or biomarkers, and the uncovering of molecular signatures associated with specific regions of the tumor.

### 3.1. Cell Types and Spatial Distribution

ST has been widely adopted to characterize the diverse cell types present in the thyroid cancer TME. For instance, Zheng et al. [22] reported greater spatial heterogeneity found in PTC samples with lymph node metastasis than those without, with major cell clusters classified as epithelial and PTC cells.

#### 3.1.1. Immune Cells

Immune cells are of particular interest when mapping distinct cell populations, as their regional expression patterns and interactions with tumor cells can help us understand tumor growth and immune evasion mechanisms, identify immunotherapy targets, and predict treatment responses. The functional capacity of these immune cells—particularly their antigen-presenting capabilities—is crucial for determining their potential to enhance the outcomes for immunotherapy like immune checkpoint inhibition.

Yan et al. [23] obtained ST data from PTC patients using 10× Genomics Visium and categorized the RNA clusters into tumor cells, normal thyroid follicular cells (FCs), atypical follicular cells (AFCs), and immune cells. By computing spatial-geneset-scores using the signature genes of each cell type to determine the regional distribution of the different cell types within the PTC samples, they observed an enrichment of B cells in spots near the center and T cells in spots near the periphery of the tumor [23]. The spatial positioning of B cells within the tumor core suggests their potential role as antigen-presenting cells, in which they could encounter tumor antigens and subsequently present them to T cells. Further assessment of these B cells’ antigen-presenting capacity, including their expressions of MHC class II molecules and co-stimulatory signals, would be critical for understanding their contribution to anti-tumor immunity and their potential to enhance cross-priming of CD8+ T cells.

Wang et al. [24] also looked into the immune cells of PTC by mapping the thyrocytes of progressive and non-progressive PTC on ST slides. They observed a higher infiltration of T and natural killer (NK) cells, particularly CD8+ T cells and regulatory T cells, in the thyrocyte clusters of progressive PTC compared to non-progressive PTC [24]. The similar spatial distributions of the two cell types throughout the progressive PTC tissue further suggested their combinatory roles in tumor immune evasion that contribute to the progression of the tumor [24]. Evaluating the expressions of antigen processing and presentation machinery genes, such as TAP1/2, PSMB8/9, and HLA class I/II genes, within these spatially defined immune cell populations could further extend our understanding into the effectiveness of CD8+ T cell responses.

For the regions with either high or low immune infiltration identified through ST, immune therapies can be tailored to target these specific areas. Future studies should incorporate functional gene signatures related to antigen presentation, T cell activation, and cross-priming efficiency to better predict which spatially defined immune microenvironments are most likely to respond to immunotherapy interventions and potentially improve treatment outcomes for thyroid cancer patients.

#### 3.1.2. Cancer-Associated Fibroblasts (CAFs)

CAFs, which are being increasingly used as targets in immunotherapy, are also important in understanding tumor progression [25,26]. Although many studies have identified CAFs in thyroid cancer using scRNA-seq [27,28,29], there are few studies on the spatial progression of CAFs. Loberg et al. [30] investigated the spatial distribution of CAFs in ATC and PTC, observing the expression of CAF markers across the tumor regions aligning with that of CDK6, a gene identified to be involved in ATC progression. Furthermore, by spatially mapping the CAF subpopulations, they observed myofibroblastic CAFs (myCAFs) adjacent to and inflammatory CAFs (iCAFs) distant from the tumor [30]. Such spatial localization also revealed a higher infiltration of myCAFs in ATC compared to PTC, indicating the contribution of myCAFs in tumor progression [30]. They further validated their results by performing immunohistochemistry on the myCAF protein POSTN and the iCAF protein APOD, demonstrating the effectiveness of integrating ST and immunohistochemistry [30].

#### 3.1.3. Colocalization

As cell types are mapped across the tumor, their spatial relationships may uncover their potential roles in the TME. For instance, after identifying T and B cells as major cell components of PTC, Tourneur et al. [31] observed distinct localization patterns of the two lymphocyte populations in the foci and epithelial areas of different PTC samples, indicating the lack of short-range communication between the two.

Moreover, Xu et al. [32] predicted the frequency of CAFs and M2 macrophages for each spatial spot from their ST data to evaluate the colocalization of the two cell populations in ATC. They observed a significant spatial correlation between the two, with a higher correlation in samples with higher Molecular Aggression and Prediction (MAP) scores [32]. As the close interaction between CAFs and M2 macrophages implies an immunosuppressive environment in ATC, especially those with high MAP scores, this suggests potential implications for immunotherapy strategies and response predictions.

#### 3.1.4. Mixed Tumors

Furthermore, for tumors mixed with different features, such as ATC, PTC, and metastatic components, the cellular compositions of different regions can be compared when spatial heterogeneity is revealed. H&E staining slides can be used to delineate the borderlines of each component so that the cell type assigned to each ST spot can be mapped to a distinct region. Using such an approach on a tissue mixed with ATC and PTC, Wang et al. [33] observed a higher concentration of myeloid cells in the ATC area, along with a uniform distribution of B, T, and NK cells in the ATC and PTC areas.

### 3.2. Gene Expression Patterns and Pathway Enrichment

Spatial context allows the expression patterns of genes and pathways to be mapped to specific histological regions, such as invasive margins or immune-infiltrated zones. By uncovering region-specific dysregulations within the TME, studies can identify key signaling pathways that drive progression to more aggressive stages, immune evasion, or treatment resistance. Such analyses offer critical information on the mechanisms that either promote or suppress the tumor development and can suggest potential biomarkers or therapeutic targets for thyroid cancer tailored to the heterogeneous characteristics of thyroid tumors. Key gene and pathway dysregulations in thyroid cancer identified by ST are summarized in Table 1.

Comparing gene expressions for different stages of thyroid cancer progression allows us to understand which genes play crucial roles in the development of the tumor. For example, Liao et al. [34] characterized distinct gene expression patterns for para-tumor (PT), PTC, locally advanced PTC (LPTC), and ATC by comparing differentially expressed genes (DEGs) analyzed from ST data [34]. In particular, PT and PTC samples showed upregulated expressions of genes such as TFF3 and SLC34A2 and exhibited low activity in most cancer hallmark pathways [34]. In contrast, LPTC and ATC samples were enriched with genes such as CXCL14 and SAA1, which were active cancer hallmarks associated with the more aggressive nature of the later stages of thyroid cancer, such as “MYC targets v1” and “G2M checkpoint” [34]. Moreover, COL7A1, LAMC2, SPHK1, and SRPX2 were chosen as biomarkers of ATC progression based on the top upregulated genes in tumor progression pathways [35]. The dysregulations of TTF1, TG, and PAX8 were also observed, as the levels of TTF1, TG, and PAX8 were more decreased in ATC than in PTC [33].

Likewise, genes and pathways enriched in invasive areas of a tumor may imply what drives the tumor cells to migrate into their surroundings. Several studies aimed to gain a deeper understanding of the invasive areas of FTC. Suzuki et al. [36] compared the ST data of the invasive area with the peripheral and central area of the tumor, detecting a subpopulation in the invasive area that expressed high levels of CD74. Condello et al. [37] looked into specific invasion areas, observing pathways like “Negative Regulation of Smooth Muscle Cell Migration” and “Collagen Fibril Organization” enriched in vascular invasive fronts and “Extracellular Matrix Assembly” and “Extracellular Matrix Organization” enriched in capsular invasive fronts. They also confirmed the dysregulations of the DPYSL3, POSTN, and TERT genes at the spots across the center and invasive front of the tumor [37].

As ST allows studies to map gene expressions based on the histologic patterns within a tissue, the squamoid, spindle, epithelioid, mixed, and inflammatory patterns of ATC have also been shown to factor into its differential gene expressions. Haq et al. [35] observed histology-specific enrichment patterns, such as the upregulation of tumorigenesis pathways in spindle, epithelioid, and mixed ATCs and the enrichment of neutrophil extracellular trap formation and NF-kappa B signaling pathways enriched only in the inflammatory group. All histologic patterns were upregulated with pathways in tight junctions and thyroid hormone synthesis, suggesting universal roles these pathways may have in the expression of ATC [35]. These findings highlight the need for a greater focus on the histological subtypes of ATC in targeting specific pathways for more personalized treatment since distinct molecular signatures and signaling pathways are involved in the ATC TME with different histological subtypes.

### 3.3. Subpopulation Diversity

The cell types identified by ST can be further clustered into subpopulations to understand how these cells behave or interact with others in the TME. Liao et al. [34] identified 16 distinct subpopulations of thyrocytes. By using spatial distributions on hexagonal lattices in 10× Genomics Visium slides to select spots surrounding each cluster of these subpopulations, they observed high proportions of B cells and T cells and thus stronger anti-tumor characteristics in clusters containing normal or pre-cancerous thyrocytes [34]. This demonstrates how the composition of the thyrocyte landscape varies as thyroid cancer progresses, specifically with regard to stronger immune infiltration, suggesting potential immunotherapy strategies.

### 3.4. Tumor Leading Edge

The tumor leading edge is the outermost region of a tumor, often characterized by distinct cellular and molecular compositions, such as immune infiltration and metabolic rate [40]. ST can reveal cell populations and interaction patterns unique to this region, which could shed light on potential metastatic dissemination, immune evasion, and therapeutic vulnerabilities. One method to do so is to identify the tumor leading edges on H&E staining images by experienced tumor pathologists and examine the cellular compositions of spatial spots along the identified region. Incorporating this, Liao et al. [34] reported a higher proportion of fibroblasts and greater immune infiltration in the leading-edge regions of ATC compared to those of PTC and LPTC [34]. They also compared the molecular differences between the leading edges and the normal regions, showing the greatest amount of gene dysregulation and suppression of signaling pathways in LPTC leading edges. Characterizing the invasive front of different types of thyroid cancer allows us to gain a fundamental understanding of the changes in the extracellular matrix and the migration tracks of tumor cells. This revelation can also support the development of therapies that specifically target tumor margins or particular cell types at the leading edge.

## 4. Tumor Evolution

Understanding how thyroid cancer evolves is important for revealing the cells of origin, driver mutations, intratumoral heterogeneity, and ultimately, therapy options through prognosis predictions. ST plays a powerful role in capturing how cells evolve within the tumor over space and time, aiding in the identification of potential biomarkers for early detection or treatment resistance, as well as uncovering therapeutic targets to prevent tumor progression.

### 4.1. Evolutionary Routes

The identification of evolutionary routes helps us track changes and predict which subclones should be of particular interest for early intervention as thyroid cancer progresses. One way to utilize spatial data to understand such routes is through spatial trajectory inference analysis. For example, Zheng et al. [22] visualized the evolutionary paths of PTC across tissue space and time by ordering the ST spots along a pseudo-space–time trajectory of tumor progression. They observed two routes toward the evolution of PTC, one where epithelial cells at the edge of the tumor tissue evolved into a cluster of PTC cells and another where PTC cells from one cluster evolved into another [22].

### 4.2. Evolution of Subpopulations

As heterogeneous as the TMEs are, it is useful to characterize the distinct roles of different cell populations in tumor evolution. Liao et al. [34] demonstrated a decrease in thyrocyte proportion across the different stages of thyroid cancer. By inferring the evolutionary trajectories of each thyrocyte subpopulation during thyroid cancer progression, they were also able to assign each subpopulation as pre-cancerous, intermediate, or late-cancerous stages [34]. The evolution pattern of thyrocytes was also observed by Yu et al. [38] through a trajectory analysis of ST data obtained from a transgenic mouse model. A cluster of thyrocytes in their final state of dedifferentiation showed upregulations of rRNA processing and translation pathways [38]. These findings have brought new insights into the evolution of thyrocytes as thyroid cancer progresses, suggesting the transition of cells and the pathways they are involved in that potentially contribute to the development of the cancer.

Moreover, Yan et al. [23] compared the involvement of FCs and AFCs during the development of tumor cells by computing pseudotime and developmental trajectories, revealing the potential role of AFCs as intermediates in the dedifferentiation of FCs into tumor cells. They also suggested a series of observations that validated the involvement of AFCs in this transition [23]. For instance, spatial–geneset scores revealed similar expression trends in hallmark gene pathways between the AFCs and the tumor cells [23]. The distance of each spatial spot from the tumor foci was also calculated to show the higher number of AFCs in regions closer to the foci [23]. Their findings therefore suggest that AFCs could serve as potential biomarkers for cancer development or contribute to finding intervention points to prevent FCs from evolving into tumor cells.

### 4.3. Invasion and Metastasis

As tumor cells evolve to be more complex and aggressive, they can participate in invasion and metastasis by colonizing new areas. The spatial variability within thyroid tumors, particularly across the benign, malignant, and metastatic areas, remains poorly understood. ST can be useful in understanding the colonizing processes, as it provides the spatial context for where the tumor cells have evolved. For example, a trajectory analysis performed on FTC samples with vascular invasion revealed that the invasive cells were further along in tumor progression [37]. These cells were enriched with genes related to epithelial–mesenchymal transition (EMT), suggesting the potential of EMT biomarkers in indicating tumor invasion [37]. Moreover, through a functional analysis on two groups of DEGs divided based on whether they were expressed in spots from a tumor tissue or a metastatic node tissue of PTC, the suppression of lipid metabolism, specifically ferroptosis, was shown to be involved in the lymph node metastasis of PTC [22]. Genes and pathways relevant to EMT and ferroptosis resistance are thus likely to act in the progression of thyroid cancer, highlighting them as therapeutic targets to prevent the spread of the tumor cells.

## 5. Cell–Cell Interactions

Comprehending interactions between cells within the TME is crucial because cellular processes such as growth and division, apoptosis, and immune suppression are regulated by such interactions [41,42]. Though scRNA-seq has been used so far to infer potential interactions, often by calculating the probability of an interaction based on the coexpression of ligands and receptors, ST limits the ligand–receptor pairs to only those that are in spatial proximity, significantly reducing the number of false positives. Key signaling pathways and molecules involved are summarized in Table 1.

### 5.1. Identification of Signaling Pathways

By comparing the crosstalk patterns of spatially co-expressed ligand–receptor pairs located in neighboring spots, Liao et al. [34] uncovered differences in the cellular interactions in PTC, LPTC, and ATC. In particular, the SERPINE1-PLAUR interaction was significantly upregulated in ATC [34]. The spatial proximity of this interaction was shown by the distributions of fibroblasts, which were highly expressed in the gene SERPINE1, and macrophages, which were highly expressed in the gene PLAUR [34]. Given that the dysregulation of the interaction was unique to ATC, the SERPINE1-PLAUR axis likely plays a role in the malignant nature of ATC and thus could act as a potential target.

Zheng et al. [22] also identified fibroblasts and macrophages to be significantly involved in the cellular interactions in PTC. They observed that, among the spots identified from their spatial evolutionary routes, the two cell types acted as major signal senders of interactions such as secretion and extracellular matrix [22]. This elucidates the roles of fibroblasts and macrophages in the tumorigenesis and evolution of PTC.

Unlike the two previous studies, Yan et al. [23] employed a unique method of drawing a contour map of the distribution of each gene across the spatial transcriptome plane to minimize the errors in analyzing ligand–receptor pathways based on manually determined trajectories. By projecting ligand–receptor pairs onto the contour plots, they identified FN1-SDC4, FN1-ITGA3, and LAMB3-ITGA2 as the top three pathways from tumor cells to AFCs to FCs [23]. Utilizing contour plots to explore communication among closely distributed cells in the PTC TME further strengthens the potential these pathways have as prognostic biomarkers or therapeutic targets of PTC.

### 5.2. Interaction Between Immune Cells and Tumor Cells

The interaction between immune cells and tumor cells is also important in understanding the tumor immune microenvironment and the roles immune cells have in tumor progression. Yan et al. [23] observed a high interaction potential between TGFβ from immune cells and TGFBR1 and TGFBR2 from tumor cells. Limited communication between APOE– tumor cells and NK, B, and T cells, especially in the CD6- and F11R-driven signaling pathways, was observed by Xiao et al. [39] through the mapping of scRNA-seq data onto spatial coordinates of PTC, which highlights the role of APOE in creating an immunosuppressive microenvironment in the tumor. Wang et al. [33] also observed immunosuppressive patterns as the tumor progresses through the stronger interactions LAMP3+ DCs had with CD8+ and regulatory T cells in the TME of progressive PTC than non-progressive PTC. Together, these interaction patterns identify cells and genes that may serve as potential therapeutic targets to reverse their immunosuppressive characteristics. Future studies can also use ST to map where within the tumor the interactions are spatially enriched to elucidate where spatially targeted therapies might be most effective.

## 6. Limitations and Future Directions

The spatial context provided by ST adds a transformative layer to our understanding of the cellular landscape of cancer. However, despite the promising potential of ST in thyroid cancer research, it still has some limitations that need to be addressed. We discuss some of the main barriers and suggest future directions to overcome them. Key points are summarized in Table 2.

### 6.1. Technical Limitations

As ST is a relatively new technology, there are several technical barriers that still need to be overcome. Key points include its low resolution, shallow sequencing depth, and limited transcriptome coverage. Sequencing-based ST platforms like 10× Genomics Visium are limited in spatial resolution, with about 5000 spots with diameters of 55 µm captured within a 6.5 mm × 6.5 mm area [18]. When applied to dense tissues like the thyroid, they cannot capture the gene expression at the single-cell level, making it difficult to distinguish different cell types in histologically heterogeneous environments. To combat this, deconvolution methods are often used to infer the cell types. For example, Xu et al. relied on a spatial deconvolution algorithm called SpaCET to detect immune subpopulations because their ST data from 10× Genomics Visium were not at single-cell resolution [32]. However, this may introduce error and bias, especially for regions with higher cell density. Although imaging-based ST like MERFISH and seqFISH can measure genes at a single-cell resolution, the number of genes that can be detected is limited to hundreds to thousands because they rely on pre-defined gene panels as detection probes [43,44]. ST technologies also capture fewer genes compared to scRNA-seq technologies, lowering their sensitivities towards rare transcripts or subtle expression changes. These limitations ultimately lead to higher costs as well as difficulties in efficient applications in clinical settings or population-scale analyses of thyroid cancer subtypes.

To enhance the data extensiveness beyond what ST alone can provide, many studies currently rely on other experimental methods and datasets, such as immunohistochemistry (IHC) and the thyroid cancer (THCA) dataset of The Cancer Genome Atlas (TCGA) database, for further validation of their findings on larger cohorts of thyroid cancer patients. Along with the need for platforms with higher sensitivities and sequencing depths, artificial intelligence (AI) models can contribute greatly to integrating ST data with different types of data to compensate for ST’s limited resolution and sensitivity. Examples of such efforts include a deep generative model that integrates spatial gene expression with histological image data to generate expression maps of higher resolution or a deep learning algorithm trained on ST data to predict gene expression based on H&E staining images [45,46]. Imputation algorithms like BayesSpace and Tangram can also enhance resolution and sensitivity by utilizing information from spatial neighborhoods and scRNA-seq data [47,48]. More accurate and sensitive deconvolution algorithms are crucial to minimizing such barriers as well.

### 6.2. Sample Preparation

Where many ST platforms require freshly frozen samples, most thyroid cancer samples are obtained in FFPE form to preserve the tissue morphology necessary for downstream pathological procedures such as H&E staining and IHC experiments [49]. Recent developments in platforms like 10× Genomics Visium and NanoString GeoMx Digital Spatial Profiling (DSP) are now compatible with FFPE samples, but they detect fewer genes and have lower sensitivities compared to the results generated by fresh-frozen tissue because RNA fragmentation by formalin causes FFPE samples to contain short and damaged RNA fragments [50,51].

Improvements in the availability of high-quality samples would allow studies involving larger cohorts and longitudinal aspects in the future, further deepening our understanding of thyroid cancer across larger populations and temporal changes.

### 6.3. Data Analysis and Interpretation

Datasets generated by ST are complex and multi-dimensional. Storing such data requires high data storage capacity, along with sophisticated analysis tools and substantial expertise to make meaningful data interpretations.

There is a need for more analysis tools and pipelines to be made publicly available for the reproducibility and collaboration of studies incorporating ST. The field will greatly benefit from AI techniques as well. For example, models that can predict which regions of the tumor have a high risk of dedifferentiation based on spatial heterogeneity would be useful for understanding which areas or genes to target. Furthermore, the integration of different types of data will also complement the limitations that ST faces. Many studies have combined ST and scRNA-seq to benefit from both the spatial and single-cell level information each technology provides. Multi-omics data, including proteomics, epigenomics, and metabolomics, are also critical for gaining a more comprehensive understanding of areas like drug targets, resistance mechanisms, and metabolic vulnerabilities. Spatial proteomics, in particular, would be useful in validating the expression patterns of ligand–receptor interactions and signaling pathways [52,53].

### 6.4. Clinical Applications

Although precision treatment of thyroid cancer would greatly benefit from clinical applications of ST, barriers including high cost, specialized equipment and expertise, and the level of complexity in analysis and interpretation limit the translation of ST into actionable diagnostics and prognostics. The lack of standardized protocols due to the technology being relatively new also makes it difficult to compare across patients and platforms.

The development of user-friendly commercial platforms or AI models that assist in identifying significant changes in expression patterns and interactions would bridge the gap to enable easier usage of the technology in clinical applications. SpatialGE, for example, is a web tool that offers a number of flexible and user-friendly functions including visualization of spatial gene expression and quantification of spatial heterogeneity [54].

Once these barriers are addressed, there are various ways to leverage ST in clinical settings. For instance, ST can complement H&E staining and IHC during diagnostic procedures, allowing for a more accurate diagnosis through a more sophisticated segmentation of the tumor boundaries or differentiation of gene expression patterns across histological regions. As ST reveals the expression of biomarkers, immune infiltration, and EMT-related pathways and genes, it can also enhance risk stratification and the identification of regions of high metastasis risk within the tumor. Furthermore, ST can be used to predict treatment responses and tailor therapy specific to individuals by offering information on the changes in spatial heterogeneity of the TME, especially in regard to the expression of the target pathways and genes, brought by treatment.

## 7. Conclusions

Studies have proven ST to be a powerful tool in providing unprecedented insight into the heterogeneity and interactions within the microenvironment of thyroid cancer. Multiple key cell populations, biomarkers, targetable pathways, and ligand–receptor interactions that potentially contribute to thyroid cancer progression and treatment resistance have been identified. However, despite rapid advances, it remains limited to widespread applications, mainly because of its complex technical barriers and high cost. Future efforts should focus on developing more accessible platforms with improved resolution, enhanced transcriptome coverage, and reduced technical complexity for routine clinical use. With these addressed, ST holds promise for more efficient and personalized diagnosis and treatment of thyroid cancer through a range of applications from uncovering biomarkers for more accurate diagnosis and risk stratification to identifying novel therapeutic targets and treatment strategies.

## Figures and Tables

**Figure 1 cells-14-00936-f001:**
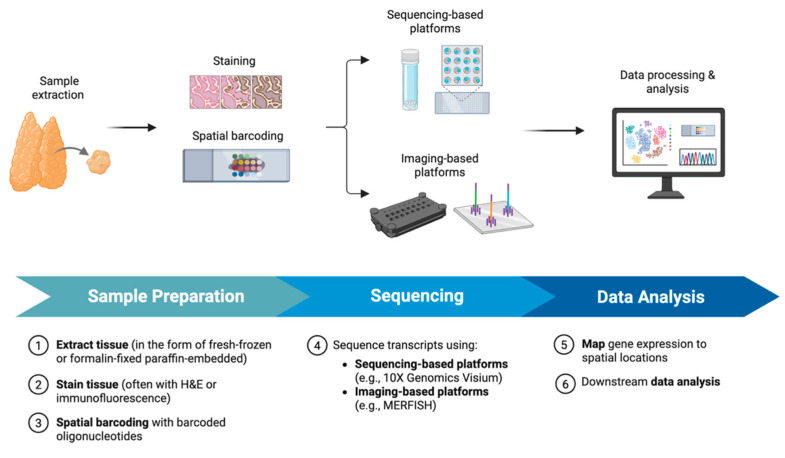
Overview of ST workflow. Samples are first extracted from the tissue in the form of fresh-frozen or FFPE. A few important steps distinguish ST from other transcriptomics technologies. Samples are stained using H&E or immunofluorescence so that the gene expression data can be mapped to their corresponding histological locations. Samples are also given spatial barcodes that contain information about where within the tissue each spot came from. These samples can be sequenced either using sequencing-based or imaging-based platforms. The main difference between the two is that sequencing-based platforms use barcoded beads while imaging-based platforms visualize spatial gene expressions directly in tissue using fluorescence or hybridization techniques. Sequenced data is then used for further downstream analysis, including cell types, tumor evolution, and cellular communication.

**Table 1 cells-14-00936-t001:** Summary of key genes, proteins, and pathways of thyroid cancer identified by ST.

Gene(s)/Pathway(s)	Observed in	Implication	Reference
*TFF3*, *SLC34A2*	PT, PTC	Low involvement in cancer pathways	Liao et al., 2025 [34]
*CXCL14*, *SAA1*	LPTC, ATC	Biomarkers of ATC progression	Liao et al., 2025 [34]
*COL7A1*, *LAMC2*, *SPHK1*, *SRPX2*	ATC		Haq et al., 2025 [35]
*TTF1*, *TG*, *PAX8*	ATC, PTC		Wang et al., 2025 [33]
*CD74*	Invasive areas of FTC	Biomarkers of FTC invasion	Suzuki et al., 2025 [36]
*DPYSL3*, *POSTN*, *TERT*, EMT-related genes	Core and invasive front of FTC		Condello et al., 2024 [37]
Negative regulation of smooth muscle cell migration; collagen-containing extracellular matrix; collagen fibril organization	Vascular invasive fronts of FTC	Mechanisms for tumor invasion, survival, and/or drug resistance at the vascular area	Condello et al., 2024 [37]
Extracellular matrix assembly; extracellular matrix organization	Capsular invasive fronts of FTC	Mechanisms for tumor invasion, survival, and/or drug resistance at the capsular area	Condello et al., 2024 [37]
rRNA processing; translation	Thyrocytes in the final dedifferentiation stage	Mechanisms thyrocytes are involved in as the tumor progresses; therapeutic targets	Yu et al., 2023 [38]
Lipid metabolism (ferroptosis)	Lymph node metastasis of PTC		Zheng et al., 2024 [22]
*SERPINE1*; *PLAUR*	ATC	Biomarker for ATC; targetable axis for ATC treatment	Liao et al., 2025 [34]
FN1-SDC4; FN1-ITGA3; LAMB3-ITGA2	PTC	Targetable axes for PTC treatment	Yan et al., 2024 [23]
TGF*β*; TGFBR1; TGFBR2	PTC	Crosstalk between immune cells and tumor cells; targets for immunotherapy	Yan et al., 2024 [23]
CD6- and F11R-driven signaling pathways	PTC		Xiao et al., 2025 [39]

**Table 2 cells-14-00936-t002:** Summary of challenges that ST currently faces and future directions for improvement.

Challenge	Description	Future Direction
Low resolution in sequencing-based platforms (e.g., 10× Genomics Visium)	Cannot reach true single-cell or subcellular resolution	Develop higher-resolution or single-cell ST platforms (e.g., Stereo-seq, CosMx); build more advanced AI models and algorithms; combine imaging and sequencing-based approaches
Limited gene detection in imaging-based platforms (e.g., MERFISH, seqFISH)	Only preselected genes are detected	
Shallow transcriptome coverage	Capture only a fraction of expressed genes	
Sample preparation	Fresh frozen tissues are often required; FFPE compatibility and sensitivity are limited in some platforms	Improve protocols for FFPE compatibility to expand clinical use
Technical complexity	Technically demanding protocols	Standardize workflows and develop user-friendly commercial kits
Data integration	ST data are complex and hard to integrate with scRNA-seq or proteomics	Advance computational tools for multi-omics integration
High cost	Need for high-quality samples, specialized equipment, high-performance storage, specialized expertise; limited scalability due to low throughput	Develop low-cost platforms; integrate with other types of data (e.g., scRNA-seq, IHC); expand community of shared datasets and training resources
Low accessibility for clinical use	Lack of reproducible, standardized pipelines	Develop user-friendly and automated software for ST data analysis and translation to clinical settings; automate ST platforms for faster clinical workflows

## Data Availability

No new data were created or analyzed in this study. Data sharing is not applicable to this article.

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
