# Peer review of "Spatial Transcriptomics in Thyroid Cancer: Applications, Limitations, and Future Perspectives"

_cells, 2025, doi:10.3390/cells14120936_

Round 1
Reviewer 1 Report
Comments and Suggestions for Authors
Excellent structure and very descriptive manuscript with the following subsections:
-Introduction
-Spatial Heterogeneity of TME
-Tumor evolution
-Cell to cell interactions
-Limitations-Future directions and conclusions.
They authors do a wonderful job in describing the heterogeneity of the TME by discussing the roles of the immune cells, stromal components such as CAFs and their co-localization and interactions.
Minor comment: When they refer to the immune cells, it would helpful if they make a point about the antigen capacity of those immune cells including B cells. Since ultimate goal is to enhance ICI or other immunotherapy outcomes, it would be important to know how functional those APCs are and how potent in cross priming CD-8 T cells could become.
Author Response
Comment: “When they refer to the immune cells, it would helpful if they make a point about the antigen capacity of those immune cells including B cells. Since ultimate goal is to enhance ICI or other immunotherapy outcomes, it would be important to know how functional those APCs are and how potent in cross priming CD-8 T cells could become.”
Response: We appreciate your insightful feedback. As suggested, we have included additional details as well as suggestions for future studies that point out how the antigen presenting capacity of B cells and cross priming of CD8+ T cells can improve immunotherapy outcomes. The added context (lines 169-171, 178-183, 190-193, and 195-199) is as follows:
“The functional capacity of these immune cells—particularly their antigen-presenting capabilities—is crucial for determining their potential to enhance the outcomes for immunotherapy like immune checkpoint inhibition.”
“The spatial positioning of B cells within the tumor core suggests their potential role as antigen-presenting cells, in which they could encounter tumor antigens and subsequently present them to T cells. Further assessment of these B cells' antigen-presenting capacity, including their expression of MHC class II molecules and co-stimulatory signals, would be critical for understanding their contribution to anti-tumor immunity and their potential to enhance cross-priming of CD8+ T cells.”
“Evaluating the expression of antigen processing and presentation machinery genes, such as TAP1/2, PSMB8/9, and HLA class I/II genes,within these spatially defined immune cell populations could further extend our understanding into the effectiveness of CD8+ T cell responses.”
“Future studies should incorporate functional gene signatures related to antigen presentation, T cell activation, and cross-priming efficiency to better predict which spatially defined immune microenvironments are most likely to respond to immunotherapy interventions and potentially improve treatment outcomes for thyroid cancer patients.”
Reviewer 2 Report
Comments and Suggestions for Authors
Spatial transcfriptomics (ST) has emerged to be a powerful tool for studying tissue biology and disease mechanisms. Thyroid cancer (TC) is an endocrine malignancy that contains 2 main histological types, i.e., papillary thyroid cancer (PTC) and anaplastic thyroid carcinoma (ATC), respectively. In this manuscript, the authors have provided a perspicacious review on how ST could be applied to examine spatial heterogeneity of the tumor microenvironment, tumor evolution, and cellular interactions. With application of ST for adding the spatial context of gene expressions that can be measured by bulk RNA sequencing (RNA-seq) and single-cell RNA-seq (scRNA-seq), molecular mechanisms of thyroid cancer progression, invasion, and metastasis could be better disentangled.
(I) Major Comments
Thyroid cancer constitutes one of the most common endocrine tumors worldwide, particularly among women (e.g., Deng Y, et al., JAMA Netw Open. 2020;3:e208759. PMID: 32589231). Although in recent year, the increasing applications of single-cell RNA-seq (scRNA-seq) have generated a wealth of information of gene expression landscape at a single-cell resolution in unraveling molecular underpinnings of thyroid cancer, scRNA-seq per se lacks spatial context because an entire tissue needs to be dissociated into individual cells before sequencing. The spatial transcriptomics (ST) technology compensates addresses this weakness by mapping transcripts across the whole tissue, thus providing spatial information of gene transcripts. Combining these scRNA-seq method and ST method could generate comprehensive and spatially resolved transcriptional profiles of a diverse range of tumor tissues. Therefore, generally speaking, the authors presented a concise and thoughtful review on several major aspects of how ST method could be employed in exploring thyroid cancer, including spatial heterogeneity of the tumor microenvironment, tumor evolution, and cell-cell interactions. However, I have several important major comments, which are shown in the following:
(1) Page 3, lines 111-113, the authors stated that
"Although many studies have identified CAFs in thyroid cancer using scRNA-seq, there are few stud-ies on the spatial progression of CAFs [23,24]."
However, Reference 25 is also a scRNA-seq study, such that the above statement could be corrected to
"Although many studies have identified CAFs in thyroid cancer using scRNA-seq [23-25], there are few studies on the spatial progression of CAFs."
(2) Page 3, lines 113-116,
"Loberg et al. investigated the spatial distri-bution of CAFs in ATC and PTC, observing the expression of CAF markers across the tumor regions aligning with that of CDK6, a gene identified to be involved in ATC pro-gression [25]."
However, the study of "Loberg et al. " is actually Reference 26, such that the above statement could be corrected to
could be corrected to
"Loberg et al. investigated the spatial distribution of CAFs in ATC and PTC, observing the expression of CAF markers across the tumor regions aligning with that of CDK6, a gene identified to be involved in ATC progression [26]."
(3) Page 4, line 132,
"in samples with higher MAP scores [28]."
could be corrected to
"in samples with higher Molecular Aggression and Prediction (MAP) scores [28]."
(4_01) In Table 1 that includes Page 7, line 133, and the entire page 8,
within Table 1, Header line, Column 1,
"Gene / Pathway"
could be corrected to
"Gene(s)/Pathway(s)"
(4_02) In Table 1 that includes Page 7, line 133, and the entire page 8,
Column 4, i.e., "Reference" Column, for all Rows of Table 1’s contents (i.e., from Row 1 to Row 14),
the authors shall add the Reference Citation Number for each and every reference cited in this Column, e.g., in Row 1, "Reference" Column,
"Liao et al., 2025"
could be corrected to
"Liao et al., 2025 [30]"
And the authors shall make corrections by adding the corresponding Reference Citation Number for every reference cited in this Column.
(5) Page 10, line 333,
"like MERFISH and seqFISH can measure genes"
could be corrected to
"like multiplexed error-robust fluorescence in situ hybridization (MERFISH) and sequential fluorescence in situ hybridization (seqFISH) can measure genes"
(6) Page 10, line 342,
"and the THCA dataset of the TCGA database,"
could be corrected to
"and the thyroid cancer (THCA) dataset of The Cancer Genome Atlas (TCGA) database,"
(7) Page 10, line 349,
"based on H&E images"
could be corrected to
"based on H&E staining images"
(8) Page 10, line 358,
"Nanostring GeoMx DSP"
could be corrected to
"NanoString GeoMx Digital Spatial Profiling (DSP)"
As shown in above corrected, "Nanostring" has been corrected to "NanoString".
(9) Page 12, lines 424-425 and the contents of the "Abbreviations" section, the authors shall add these following into this section,
"DSP Digital Spatial Profiling"
"IHC Immunohistochemistry"
"MAP Molecular Aggression and Prediction"
"MERFISH Multiplexed error-robust fluorescence in situ hybridization"
"RNA-seq RNA sequencing"
"SeqFISH Sequential fluorescence in situ hybridization"
"TCGA The Cancer Genome Atlas"
"THCA Thyroid cancer"
After adding these above terms, the authors shall sort all these terms in the "Abbreviations" section in alphabetical order for ease of reference.
(10) Page 15, lines 570-572,
"Ospina, O., Manjarres-Betancur, R., Gonzalez-Calderon, G., Soupir, A. C., Smalley, I., Tsai, K., Markowitz, J., Berglund, A., Yu, X., & Fridley, B. L. (2024). spatialGE: Empowering researchers to study the tumor microenvironment leveraging spatial tran-scriptomics. Cancer Research, Abstract 876. https://doi.org/10.1158/1538-7445.AM2024-876"
could be corrected to
"Ospina, O., Manjarres-Betancur, R., Gonzalez-Calderon, G., Soupir, A. C., Smalley, I., Tsai, K. Y., Markowitz, J., Khaled, M. L., Vallebuona, E., Berglund, A. E., Eschrich, S. A., Yu, X., & Fridley, B. L. (2025). spatialGE Is a User-Friendly Web Application That Facilitates Spatial Transcriptomics Data Analysis. Cancer Research, 85(5), 848-858. https://doi.org/10.1158/0008-5472.CAN-24-2346"
The above just shows 1 example, and for "References" section that ranges from Page 12, line 426, to Page 15, line 572,
the authors shall thoroughly check the accuracy and completeness of all references. and in the main text that consists of 6 sections that include Table 1 and Table 2 and Figure 1, the authors shall double check that these references are correctly cited
(II) Minor Comments
First of all, in the main text that consists of 6 sections that include Table 1 and Table 2 and Figure 1,
all occurrences of "H&E images" shall be corrected to "H&E staining images", to use the standard term "H&E staining images" consistently,
all occurrences of "co-localization" shall be corrected to "colocalization", to use the standard term "colocalization" consistently,
all occurrences of "10X Visium" shall be corrected to "10X Genomics Visium", to use the standard term "10X Genomics Visium" consistently,
all occurrences of "10x Visium" shall be corrected to "10X Genomics Visium", to use the standard term "10X Genomics Visium" consistently, and
all occurrences of "Nanostring" shall be corrected to "NanoString", to use the standard term "NanoString" consistently.
In addition, there are a variety of grammatical and typographical errors that should be corrected, which are indicated in the following:
(1) Page 2, line 43,
"RNA sequencing, in"
could be corrected to
"RNA sequencing (RNA-seq), in"
(2) Page 2, line 44,
"has provided valuable insights into the diversity"
could be corrected to
"has provided valuable information into the diversity"
(3) Page 2, line 45,
"by offering insight into the differences"
could be corrected to
"by offering profound insights into the differences"
(4) Page 2, line 47,
"by analyzing single-cell RNA sequencing (scRNA-seq) data"
could be corrected to
"by analyzing single-cell RNA-seq (scRNA-seq) data"
(5) Page 2, lines 48-50,
"have also elucidated valuable information about the cancer, such as the critical roles of CADM1_CADM1 ligand recep-tor in the regulation of PTC"
could be corrected to
"have also elucidated intricate molecular mechanisms about the cancer, such as the critical roles of CADM1-CADM1 interactions in the regulation of PTC"
(6) Page 2, line 51,
"However, both bulk and scRNA-seq"
could be corrected to
"However, both bulk RNA-seq and scRNA-seq"
(7) Page 2, line 53,
"To overcome this, spatial transcriptomics (ST)"
could be corrected to
"To overcome such an impediment, spatial transcriptomics (ST)"
(8) Page 2, line 55,
"gene activity to specific locations."
could be corrected to
"gene expressions to specific locations."
(9) Page 2, line 56,
"it captures mRNA directly from slides or beads"
could be corrected to
"it captures mRNA molecules directly from either slides or beads"
(10) Page 2, line 57,
"It offers critical insight into the"
could be corrected to
"Further, it provides critical insights into the"
As shown in above corrected, "insight" has been corrected to "insights".
(11) Page 2, line 62,
"a patient's tumor microenvironment. The histologically"
could be corrected to
"a patient's tumor microenvironment (TME). The histologically"
(12) Page 2, line 67,
"advancing its application."
could be corrected to
"advancing its applications."
(13) Page 2, line 68,
"of the technology"
could be corrected to
"of this burgenoning technology"
(14) Page 2, line 72,
"Spatial Heterogeneity of the Tumor Microenvironment"
could be corrected to
"Spatial Heterogeneity of the Tumor Microenvironment (TME)"
(15) Page 2, line 73,
"The tumor microenvironment (TME) refers"
could be corrected to
"The TME refers"
(16) Page 2, line 74,
"like cancer-associated fibroblasts and"
could be corrected to
"like cancer-associated fibroblasts (CAFs), and"
(17) Page 3, line 74,
"like cancer-associated fibroblasts and"
could be corrected to
"like cancer-associated fibroblasts (CAFs), and"
(18) Page 3, line 78,
"and communication within the environment"
could be corrected to
"and their communications within the environment"
(19) Page 3, line 96,
"atypical thyroid follicular cells (AFCs),"
could be corrected to
"atypical follicular cells (AFCs),"
(20) Page 3, line 102,
"and NK cells, particularly"
could be corrected to
"and natural killer (NK) cells, particularly"
(21) Page 3, line 106,
"With the regions with high or low immune infiltration"
could be corrected to
"For the regions with either high or low immune infiltration"
(22) Page 3, line 109,
"2.1.2. Cancer-Associated Fibroblasts"
could be corrected to
"2.1.2. Cancer-Associated Fibroblasts (CAFs)"
(23) Page 3, line 110,
"Cancer-associated fibroblasts (CAFs) are also important"
could be corrected to
"CAFs are also important"
(24) Page 3, lines 110-111,
"pro-gression and are being increasingly used as targets"
could be corrected to
"progression, which are being increasingly used as targets"
(25) Page 4, line 130,
"to evaluate the co-localization of"
could be corrected to
"to evaluate the colocalization of"
(26) Page 4, lines 150-151,
"that drive or suppress the tumor and can"
could be corrected to
"that either promote or suppress the tumor development and can"
(27) Page 4, line 154,
"play a role in the development"
could be corrected to
"play crucial roles in the development"
(28) Page 4, line 160,
"which were active in cancer hallmarks"
could be corrected to
"which were active cancer hallmarks"
(29) Page 4, line 164,
"dysregulation of TTF1, TG, and PAX8 was"
could be corrected to
"dysregulations of TTF1, TG, and PAX8 were"
(30) Page 5, lines 174-175,
"the dysregulation of the DPYSL3, POSTN, and TERT genes"
could be corrected to
"the dysregulations of the DPYSL3, POSTN, and TERT genes"
(31) Page 5, line 177,
"to map gene expression based on"
could be corrected to
"to map gene expressions based on"
As shown in above corrected, "expression" has been corrected to "expressions".
(32) Page 5, line 179,
"factor into its differential gene expression. Haq et al. observed"
could be corrected to
"factor into the differential gene expressions. Haq et al. [31] observed"
As shown in above corrected, "expression" has been corrected to "expressions".
(33) Page 5, line 191,
"Liao et al. identified"
could be corrected to
"Liao et al. [30] identified"
(34) Page 5, line 193,
"in Visium slides to select spots surrounding each"
could be corrected to
"in 10X Genomics Visium slides to select spots surrounding each"
(35) Page 5, line 203,
"allowing insight into potential metastatic dissemination"
could be corrected to
"which could shed light on potential metastatic dissemination"
(36) Page 5, line 206,
"Liao et al. reported a higher"
could be corrected to
"Liao et al. [30] reported a higher"
(37) Page 5, line 211,
"of different types of thyroid cancer lets us understand the"
could be corrected to
"of different types of thyroid cancer allows us to gain a fundamental understanding of the"
(38) Page 6, line 226,
"Zheng et al. visualized the evolutionary paths"
could be corrected to
"Zheng et al. [18] visualized the evolutionary paths"
(39) Page 6, line 233,
"Liao et al. demonstrated a decrease"
could be corrected to
"Liao et al. [30] demonstrated a decrease"
(40) Page 6, line 237,
"was also observed by Yu et al. through"
could be corrected to
"was also observed by Yu et al. [35] through"
(41) Page 6, line 239,
"showed upregulation of rRNA"
could be corrected to
"showed upregulations of rRNA"
(42) Page 6, line 240,
"These bring insight into"
could be corrected to
"These findings have brought new insights into"
(43) Page 6, line 243,
"Moreover, Yan et al. compared the involvement"
could be corrected to
"Moreover, Yan et al. [19] compared the involvement"
(44) Page 7, line 278,
"Liao et al. uncovered differences"
could be corrected to
"Liao et al. [30] uncovered differences"
(45) Page 7, line 285,
"Zheng et al. also identified fibroblasts and macrophages"
could be corrected to
"Zheng et al. [18] also identified fibroblasts and macrophages"
(46) Page 7, line 287,
"their spatial evolutionary route, the two cell types"
could be corrected to
"their spatial evolutionary routes, the two cell types"
As shown in above corrected, "route" has been corrected to "routes".
(47) Page 7, line 290,
"Yan et al. employed a unique method"
could be corrected to
"Yan et al. [19] employed a unique method"
(48) Page 9, line 320,
"Summary of challenges ST currently faces and future directions to improve it."
could be corrected to
"Summary of challenges that ST currently faces and future directions for improvement."
(49) Page 10, line 324,
"like 10X Visium are"
could be corrected to
"like 10X Genomics Visium are"
(50) Page 10, line 331,
"ST data from 10X Visium was not single-cell resolution"
could be corrected to
"ST data from 10X Genomics Visium were not at single-cell resolution"
(51) Page 10, line 336,
"lowering their sensitivity towards rare"
could be corrected to
"lowering their sensitivities towards rare"
(52) Page 10, line 344,
"with higher sensitivity and sequencing depth,"
could be corrected to
"with higher sensitivities and sequencing depths,"
As shown in above corrected, "sensitivity" has been corrected to "sensitivities", and "depth" has been corrected to "depths", respectively
(53) Page 10, line 357,
"in platforms like 10x Visium and"
could be corrected to
"in platforms like 10X Genomics Visium and"
(54) Page 10, line 359,
"have low sensitivity compare to"
could be corrected to
"have low sensitivities compare to"
(55) Page 10, line 366-367,
"re-quires high computational capacity, along with sophisticated analysis tools and expertise"
could be corrected to
"requires s high data storage capacity, along with sophisticated analysis tools and substantial expertise"
(56) Page 11, line 374,
"the limitations ST faces."
could be corrected to
"the limitations that ST faces."
(57) Page 11, line 377,
"is also critical in gaining a more comprehensive understanding of"
could be corrected to
"are also critical for gaining a more comprehensive understanding of"
(58) Page 11, line 390,
"gap to enable easier use of the technology"
could be corrected to
"gap to enable easier usage of the technology"
The above are just several examples, and the authors shall perform a careful and thorough checking on the main text to correct all errors.
Comments on the Quality of English Language
Moderate editing of English language is required, as specified in both the Major Comments and Minor Comments for the authors
Author Response
We greatly appreciate your detailed feedback. We have corrected all things pointed out and have further revised the whole draft based on your suggestions.
1) As suggested by comment 4, we have added the Reference Citation Number in the reference column for table 1 (page 9, line 409).
2) As suggested by comment 9, we have updated the abbreviations table (page 14, line 528) so that it includes DSP, IHC, MAP, MERFISH, RNA-seq, SeqFISH, TCGA, and THCA. We also ordered the rows alphabetically.
3) As suggested by comment 10, we have checked all of our references to ensure that they cite our sources appropriately.
4) We have improved our English language throughout our draft to improve clarity and flow. For instance:
- This leads the way for more accurate diagnoses and personalized therapies, where treatment plans can be tailored based on the unique spatial characteristics" was corrected to "ST thus paves the way for more accurate diagnoses and personalized treatment plans tailored to the unique spatial characteristics" in line 64.
- "we hope to gather a collective understanding" was corrected to "we hope to foster a deeper understanding" in line 74.
- "Due to the critical role the TME has in tumor initiation" was corrected to "Due to its pivotal role in tumor initiation" in line 149.
- "expression" was corrected to "expressions" in lines 156 and 250.
- "Many studies have utilized ST to identify the different cell types within the TME of thyroid cancer." was corrected to "ST has been widely adopted to characterize the diverse cell types present in the thyroid cancer TME." in line 160.
- "observed by how the levels of TTF1, TG, and PAX8 were more decreased in ATC than PTC" was corrected to "observed, as the levels of TTF1, TG, and PAX8 were more decreased in ATC than in PTC" in line 256.
- "to understand how the cells behave or interact with other cells" was corrected to "to understand how these cells behave or interact with others" in line 283.
- "pointing them as" was corrected to "highlighting them as" in line 361.
- "suggest cells and genes that may act as potential targets in therapy" was corrected to "identify cells and genes that may serve as potential therapeutic targets" in line 404.
- "where spatially targeted therapies might need to be focused" was corrected to "where spatially targeted therapies might be most effective" in line 408.
5) We moved the in-text reference numbers to come after the authors' names, as suggested by your comments 36, 39, 40, 43, 44, 45, and 47. These can be found in line 162 (Zheng et al. [22]), line 172 (Yan et al. [23]), line 184 (Wang et al. [24]), line 217 (Tourneur et al. [31]), line 220 (Xu et al. [32]), line 233 (Wang et al. [33]), line 247 (Liao et al. [34]), line 261 (Suzuki et al. [36]), line 263 (Condello et al. [37]), line 397 (Yan et al. [23]), line 400 (Xiao et al. [42]), and line 402 (Wang et al. [33]).
Reviewer 3 Report
Comments and Suggestions for Authors
The authors present a study on the use of ST in thyroid cancer research, addressing both its
recent applications and current limitations, as well as potential future directions. The
manuscript compiles a considerable number of relevant studies.
Below are some comments and suggestions for the authors:
1. The introduction appropriately begins by presenting the clinical context of thyroid
cancer; however, the transition to the discussion on next-generation sequencing
technologies is somewhat abrupt. I suggest establishing a smoother connection
between these sections by explaining, for example, how the limitations of current
therapies drive the need for more advanced technologies to characterize tumors at
the molecular level.
2. Figure 1 is visually well-constructed, but it is not referenced in the body of the text.
Furthermore, its content may be complex for readers who are not familiar with the
technology. I recommend adding an explanatory paragraph to accompany the figure,
clearly describing what is being represented.
3. Although the article centers on ST, it does not include a sufficiently detailed
description of what this technology entails, how it works, and what advantages it
offers over other transcriptomic techniques. This gap is particularly noticeable given
that it is a relatively new tool for many readers. I suggest including a brief initial
section or an introductory paragraph (ideally linked to Figure 1) that clearly and
accessibly explains the basic principles of ST, the technical characteristics of the
tissues and the technology, as well as its main platforms.
Author Response
Comment 1: “The introduction appropriately begins by presenting the clinical context of thyroid cancer; however, the transition to the discussion on next-generation sequencing technologies is somewhat abrupt. I suggest establishing a smoother connection between these sections by explaining, for example, how the limitations of current therapies drive the need for more advanced technologies to characterize tumors at the molecular level.”
Response 1: Thank you for pointing this out. As recommended, we have added the following in lines 40-45: “These current therapies, however, lack precise targeting of specific mutations and pathways and are limited in addressing variable clinical responses due to tumor hetereogeneity. This has highlighted the need for more advanced technologies to better characterize tumors at the molecular level. Next-generation sequencing technologies offer a powerful means to uncover the genomic and transcriptomic alterations driving disease progression”
Comment 2: “Figure 1 is visually well-constructed, but it is not referenced in the body of the text. Furthermore, its content may be complex for readers who are not familiar with the technology. I recommend adding an explanatory paragraph to accompany the figure, clearly describing what is being represented.”
Response 2: We appreciate your constructive suggestion. We have, accordingly, modified Figure 1 and expanded its caption so that the information would be more easily accessible to the readers. They can be found in lines 136-146:
“Figure 1. Overview of ST workflow. Samples are first extracted from the tissue in the form of fresh-frozen or FFPE. A few important steps distinguish ST from other transcriptomics technologies. Samples are stained using H&E or immunofluorescence so that the gene expression data can be mapped to their corresponding histological locations. Samples are also given spatial barcodes that contain information about where within the tissue each spot came from. These samples can be sequenced either using sequencing-based or imaging-based platforms. The main difference between the two is that sequencing-based platforms use barcoded beads while imaging-based platforms visualize spatial gene expression directly in tissue using fluorescence or hybridization techniques. Sequenced data is then used for further downstream analysis, including cell types, tumor evolution, and cellular communication.”
Comment 3: “Although the article centers on ST, it does not include a sufficiently detailed description of what this technology entails, how it works, and what advantages it offers over other transcriptomic techniques. This gap is particularly noticeable given that it is a relatively new tool for many readers. I suggest including a brief initial section or an introductory paragraph (ideally linked to Figure 1) that clearly and accessibly explains the basic principles of ST, the technical characteristics of the tissues and the technology, as well as its main platforms.”
Response 3: Thank you for your valuable feedback. We agree that the paper does not fully provide the details of how spatial transcriptomics works, and have therefore added a brief section that addresses the advantages of the technology, its workflow, and its popular platforms. The revised section (lines 78-135) is as follows:
“2. Overview of ST
ST is an innovative class of technologies that enables the measurement of gene expression while retaining the spatial organization of cells within intact tissue sections. Unlike bulk RNA-seq, which averages gene expression across heterogeneous cell populations, or scRNA-seq, which requires tissue dissociation and sacrifices spatial context, ST preserves the tissue architecture and allows researchers to map transcripts back to their original histological locations. By integrating molecular profiling with spatial context, ST bridges histopathology and transcriptomics, allowing a range of functions like dissecting tumor heterogeneity, mapping tumor-immune interactions, and identifying niche-specific drug resistance mechanisms. To aid in the understanding of how ST is used in thyroid cancer research, we first briefly provide an overview of the workflow of ST and the common platforms used (Figure 1).
2.1. Workflow and Methodology of ST
ST begins by preparing tissue sections, typically fresh-frozen or formalin-fixed paraffin-embedded (FFPE), mounted onto specialized slides containing spatially barcoded capture probes. Tissue sections are stained—commonly with hematoxylin and eosin (H&E) or immunofluorescence—for histological reference, followed by high-resolution imaging. This image serves as a spatial map to which transcriptomic data are later aligned.
After imaging, the tissue is permeabilized to release mRNA molecules, which hybridize to the barcoded oligonucleotides in a position-dependent manner. These oligonucleotides include a spatial barcode unique to each capture spot or bead, a poly(dT) sequence to bind polyadenylated transcripts, and a unique molecular identifier for transcript quantification. Captured transcripts are then reverse-transcribed, amplified, and subjected to high-throughput sequencing. The resulting data are computationally processed to map gene expression back to precise spatial locations within the tissue [17].
The resolution and sensitivity of ST depend on several experimental variables, including tissue quality, thickness, permeabilization conditions, and RNA integrity. ST data are often integrated with histological features and other data types like proteomics or scRNA-seq to provide a multimodal understanding of tissue architecture and cellular interactions. In the context of thyroid cancer, this workflow enables the spatial mapping of oncogenic gene expression patterns, immune infiltration, and microenvironmental heterogeneity, offering insights beyond what bulk or dissociated single-cell approaches can achieve.
2.2. ST Platforms
There are various platforms available for performing ST, each differing in resolution, transcript coverage, and technical requirements. These platforms can be broadly categorized into sequencing-based and imaging-based methods.
Sequencing-based platforms, such as 10x Genomics Visium, Slide-seq, and Slide-seqV2, use arrays or beads embedded with spatial barcodes to capture mRNA. 10x Visium, for example, uses capture spots approximately 55 µm in diameter that can cover multiple cells per spot, enabling whole-transcriptome analysis with moderate resolution [18]. Slide-seq and Slide-seqV2 can improve spatial resolution to about 10 µm by using randomly distributed barcoded beads on a slide, approaching single-cell granularity [15]. These platforms offer unbiased transcriptome-wide coverage and are well-suited for fresh-frozen tissues.
Imaging-based platforms, such as MERFISH (Multiplexed Error-Robust FISH), seqFISH, and CosMx SMI, use rounds of hybridization and high-resolution imaging to detect hundreds to thousands of pre-selected RNA targets at subcellular resolution [19, 20]. While these methods offer greater spatial precision than sequencing-based platforms, they are typically limited to targeted panels and require more complex imaging infrastructure and analysis pipelines [17].
Each platform presents trade-offs in resolution, transcriptome coverage, scalability, and sample type compatibility. Visium, for instance, is ideal for broad discovery in research settings, whereas CosMx may be more applicable for targeted clinical assays. As ST technology continues to evolve, improvements in resolution, sensitivity, and multimodal integration are making it increasingly accessible and powerful for clinical research in thyroid cancer.”
Round 2
Reviewer 2 Report
Comments and Suggestions for Authors
In this revised manuscript, Revision 1 (i.e., R1), the authors have addressed a majority of reviewer comments, but there are the following further comments that the authors shall address adequately:
(1) Page 3 line 115,
"such as 10x Genomics Visium,"
could be corrected to
"such as 10X Genomics Visium,"
(2) Page 3, line 117,
"10x Visium, for example,"
could be corrected to
"10X Genomics Visium, for example,"
(3) Page 3, lines 123-124,
"such as MERFISH (Multiplexed Error-Robust FISH), seqFISH, and CosMx SMI,"
could be corrected to
"such as multiplexed error-robust fluorescence in situ hybridization (MERFISH), sequential fluorescence in situ hybridization (seqFISH), and CosMx Spatial Molecular Imager (SMI),"
(4_01) Page 4, top section, inside Figure 1,
In "Sequencing" part, under "Sequencing-based platforms",
"e.g. 10X Visium"
could be corrected to
"e.g., 10X Genomics Visium"
As shown in above corrected, "e.g." has been corrected to "e.g.", and "10X Visium" has been corrected to "10X Genomics Visium", respectively
(4_02) Page 4, top section, inside Figure 1,
In "Sequencing" part, under "Imaging-based platforms",
"e.g. MERFISH"
could be corrected to
"e.g., MERFISH"
As shown in above corrected, "e.g." has been corrected to "e.g.,"
(5_01) Page 11, within Table 2, Row 1, Column 1 (i.e., "Challenge" Column),
"10X Visium"
could be corrected to
"10X Genomics Visium"
(5_02) Page 11, within Table 2, Row 2, Column 1 (i.e., "Challenge" Column),
" e.g. MERFISH, seqFISH"
could be corrected to
" e.g., MERFISH, seqFISH"
As shown in above corrected, "e.g." has been corrected to "e.g.,"
(6) Page 12, lines 430-432,
"Although imaging-based ST like multiplexed error-robust fluorescence in situ hybridization (MERFISH) and sequential fluorescence in situ hybridization (seqFISH) can measure"
could be corrected to
"Although imaging-based ST like MERFISH and seqFISH can measure"
(7) Page 14, in "Abbreviations" section,
"SeqFISH Sequential fluorescence in situ hybridization"
could be corrected to
"SeqFISH Sequential fluorescence in situ hybridization
SMI Spatial Molecular Imager"
Several editings of English language is required, as specified in these further Comments for the authors
Author Response
Thank you once again for your detailed comments. We have revised the draft accordingly. Specifically:
(1) "such as 10x Genomics Visium," (Page 3 line 115) has been corrected to "such as 10X Genomics Visium,"
(2) "10x Visium, for example," (Page 3, line 117) has been corrected to "10X Genomics Visium, for example,"
(3) "such as MERFISH (Multiplexed Error-Robust FISH), seqFISH, and CosMx SMI," (Page 3, lines 123-124) has been corrected to "such as multiplexed error-robust fluorescence in situ hybridization (MERFISH), sequential fluorescence in situ hybridization (seqFISH), and CosMx Spatial Molecular Imager (SMI),"
(4_01) "e.g. 10X Visium" (Figure 1, under "Sequencing-based platforms") has been corrected to "e.g., 10X Genomics Visium"
(4_02) "e.g. MERFISH" (Figure 1, under "Sequencing-based platforms") has been corrected to "e.g., MERFISH"
(5_01) "10X Visium" (Table 2, Row 1, Column 1) has been corrected to "10X Genomics Visium"
(5_02) " e.g. MERFISH, seqFISH" (Table 2, Row 2, Column 1) has been corrected to " e.g., MERFISH, seqFISH"
(6) "Although imaging-based ST like multiplexed error-robust fluorescence in situ hybridization (MERFISH) and sequential fluorescence in situ hybridization (seqFISH) can measure" (Page 12, lines 430-432) has been corrected to "Although imaging-based ST like MERFISH and seqFISH can measure"
(7) "SeqFISH Sequential fluorescence in situ hybridization" (Page 14, "Abbreviations" section) has been corrected to "SeqFISH Sequential fluorescence in situ hybridization
SMI Spatial Molecular Imager".
Reviewer 3 Report
Comments and Suggestions for Authors
The authors made the changes suggested by the reviewer to the manuscript.
Author Response
We greatly appreciate your valuable input on our draft.